# How Do Nitrogen Deposition, Mowing, and Deer Grazing Drive Vegetation Changes on Dune Heaths?

Mathias Emil Kaae [1], Fenjuan Hu [2], Jesper Leth Bak [3], Morten Tune Strandberg [3] and Christian Frølund Damgaard [3,*]

1   Department of Ecoscience, Section of Terrestrial Ecology, Aarhus University, 1110, C. F. Møllers Allé 8, 8000 Aarhus, Denmark; mek@ecos.au.dk
2   VIA University College, Forskningscenter for byggeri, klima, vandteknologi og digitalisering, Banegårdsgade 2, 8700 Horsens, Denmark; fehu@via.dk
3   Department of Ecoscience, Section of Terrestrial Ecology, Aarhus University, 1120, C. F. Møllers Allé 4, 8000 Aarhus, Denmark; jlb@ecos.au.dk (J.L.B.); mts@ecos.au.dk (M.T.S.)
*   Correspondence: cfd@ecos.au.dk; Tel.: +45-30-18-31-53

**Abstract:** Heathland vegetation has undergone significant changes in the past century, e.g., due to airborne pollutants and a lack of proper management. Understanding the interactions between these factors in combination is pivotal for heathland conservation. Here, we studied the vegetation changes at a dune heath in a four-year manipulation experiment analysing the combined effects of nitrogen deposition, mowing, and deer grazing. Our results showed no significant effect of nitrogen deposition and deer grazing on plant growth and cover of dwarf shrubs within the experimental plots. However, high loads of nitrogen decreased bryophyte cover and increased the growth and cover of sand sedge *Carex arenaria* L. Mowing adversely affected the dwarf shrub community, e.g., the dwarf shrub species crowberry *Empetrum nigrum* L., and facilitated increased cover and plant growth of graminoids. Plant growth and the cover of *C. arenaria* increased in plots without deer grazing, whereas bryophyte cover decreased significantly without grazing. We do not recommend intensive mowing of vegetation as a conservation method for dune heaths because it promotes graminoids. From a conservation aspect, it is essential to consider the effect of deer on heathlands because they both impede some species and benefit others and mitigate the adverse effects of nitrogen deposition on dune heaths.

**Keywords:** heathland conservation; dwarf shrubs; graminoid expansion; bryophytes; management; manipulation experiment



## 1. Introduction

Decalcified fixed dunes with *Empetrum nigrum* and other dwarf shrubs (Annex I habitat type: 2140) [1], hereafter termed dune heaths, cover significant land areas along the west coast of Jutland in Denmark [2–4]. Dune heaths are a habitat for a diverse range of endangered and rare species, making them a critical conservation target [5,6]. According to the Habitat Directive (92/43/CEE), dune heaths are a habitat of priority importance where the conservation status in Denmark is unfavourable/inadequate in the Atlantic and Continental regions [7]. Field studies from nutrient-poor habitats exposed to excess nitrogen confirm that more nitrogen gives an observable change in vegetation and the soil e.g., [8–10]. As a management tool, mowing and removing the material or grazing the habitat may counteract excess nitrogen on heathlands; however, data indicate that it removes less nitrogen than other conservation methods (e.g., sod-cutting) [11]. While several manipulation studies address the effects of excess nitrogen and mowing on heathland habitats e.g., [9,11–13], fewer studies examine the effects of deer grazing on heathlands e.g., [14,15]. Here, we examine how nitrogen deposition, mowing, and

deer grazing affect dune heath vegetation because, to our knowledge, no studies have investigated how all factors combined impact dune heath vegetation.

Especially from the 1950s to today, Northwestern Europe's heathlands have presented increased amounts of reactive nitrogen [8,16–19]. It is an environmental crisis that keeps challenging the conservation status of heathlands [19–23]. Meanwhile, there have been significant advancements in our knowledge of the effects of nitrogen on vegetation facilitated through nitrogen manipulation experiments e.g., [9,24]. Manipulation experiments are used to estimate empirical critical nitrogen loads ($CL_{emp}N$) for habitats, including dune heaths [17]. They are essential for evaluating the effects of nitrogen and are, in particular, valuable when conducted in areas with low nitrogen deposition because they may be used to simulate realistic deposition levels here [17]. In Denmark, most reduced nitrogen, such as ammonia, comes from animal husbandry and most (20 to 60%) is deposited within two kilometres of the source [25]. Most nitrogen is stored below ground, with less than one-third deposited in the plant biomass, i.e., in the litter or living material [24].

Long-term effects of nitrogen deposition include amendments in species composition, which on heathlands means replacing dwarf shrubs, lichens, bryophytes, and forbs with graminoids and increasing disturbance and stress factors [8,9,26]. For habitats in general, the most evident change is the reduced plant species richness and enhanced succession [8,17,27]. In the short term, increased nitrogen affects plant vegetation by increasing litter production, the shoot/root ratio, nitrogen uptake and plant availability, and mineralisation [8,28,29]. In soil, increased nitrogen deposition can lower ANC and thereby reduce the nitrification process, leading to, among other things, higher ammonium/nitrate ratios, causing litter accumulation and increasing the amount of toxic metals such as Al and Fe [8]. The higher ammonium/nitrate ratio coincides with a decline in rare vascular plants on heathlands [6]. Moreover, excess nitrogen reduces the number of base cations lost due to leaching [8]. According to Aerts and Bobbink [8], Power, et al. [30,31], nitrogen deposition amplifies disturbance factors such as herbivory from the heather beetle *Lochmaea suturalis* (Thomson, 1866) or increases the impact of stress factors such as frost and drought; these results are corroborated in the review in [32]. The invasion of *L. suturalis* may open the heather *Calluna vulgaris* (L.) Hull canopy and facilitate the transition from dwarf shrub-dominated heathland towards grassland [12,30]. However, some findings broaden this view and underline that if *C. vulgaris* can maintain a closed canopy, it often wins the competition for space over graminoids, at least for a while [33]. The entire life cycle of *C. vulgaris* is affected by nitrogen deposition, where some results indicate that it may have a shorter lifespan and higher productivity [24,34].

Bähring, et al. [9] reported that for other species, a low nitrogen deposition level could cause amendments in the vegetation if the critical load of reactive nitrogen is exceeded chronically. They showed increased coverage of graminoids (57–71 per cent) at $\geq$10 kg N ha$^{-1}$ year$^{-1}$ and a decrease in the coverage of bryophytes and lichens, with deposition levels above 10 kg N ha$^{-1}$ year$^{-1}$. However, *C. vulgaris* has a minor decrease in coverage at high doses $\geq$ 50 kg N ha$^{-1}$ year$^{-1}$ in the three years their experiment lasted. The expansion of graminoids' occurrence also depends on the site's initial soil condition; here, Remke, et al. [35] found that on dune heaths, sand-sedge *Carex arenaria* L. populations expanded at non-calcareous sites when exposed to excess nitrogen, in contrast to no increase in *C. arenaria* at calcareous sites. However, nitrogen deposition is not the only source of amendments to vegetation composition on heathlands; management also plays a crucial role [8].

Proper management may preserve and counteract the adverse effects of nitrogen deposition [36]; however, according to [37], nature conservation measures should be versatile. Also, Schellenberg and Bergmeier [37] underline the essential effects of disturbance. Their findings show that present-day floristic diversity is related to the early successional stages and links the growth stages of *C. vulgaris* to species diversity and composition. Härdtle, et al. [11] studied the effects of various management interventions on heathlands to counteract excess nitrogen. They concluded that some management interventions (e.g., sod-cutting

or prescribed burning) could mitigate the adverse effects of nitrogen deposition. However, preserving heathlands without high-intensity management, such as sod-cutting, might prove challenging because grazing, mowing, and prescribed burning may be insufficient in removing excess nitrogen [11,24,36,38]. Additionally, the evidence from mowing experiments corroborates the proposition from Schellenberg and Bergmeier [37] that mowing homogenises vegetation and reduces arthropod diversity [39–41]. Two studies from the United Kingdom, Britton, et al. [42,43], point out that mowing or cutting did not increase the cover of *C. vulgaris.* However, results from Milligan, et al. [43] state that mowing increases species diversity related to bare soil and reduces the cover of purple moor-grass *Molinia caerulea* (L.) Moench.

Historically, domesticated ungulates grazed European heathlands, including cattle, horses, ponies, sheep, and goats. In Denmark, sheep grazed heathlands all year round, while cattle were often transported to the stables to collect their dung for fertilising fields; however, these traditional livestock farming practices have disappeared in most places [44]. In contrast, the number of deer in Denmark has been increasing in the past decades, which means that they have a significant effect on vegetation, but this may not be comparable effect to sheep and cattle due to their different feeding habits [15,45,46]. Other substantial differences exist between deer and domesticated ungulates, such as cattle and ponies; cattle and ponies have larger hooves than sheep and deer, meaning they affect the vegetation differently. The literature highlights that reducing the number of livestock at a heathland benefits the dwarf shrub vegetation while excluding cattle, sheep, and red deer entirely might be favourable [47,48]. Hartley and Mitchell [48] mentioned that more nutrient-rich vegetation attracts more grazers, which increases the amount of dung and urine deposited and the frequency of trampling on vegetation; they also mentioned that plots without grazing but where nitrogen was added (75 kg N ha$^{-1}$ year$^{-1}$) did not experience a decline in *C. vulgaris* cover. The grazers in our study are divided into two separate feeding groups, where roe deer *Capreolus capreolus* (Linnaeus, 1758) is a concentrate selector (i.e., a browser) and red deer *Cervus elaphus* (Linnaeus, 1758) and fallow deer *Dama dama* (Linnaeus, 1758) are intermediate feeders (i.e., both browsers and grazers). Meanwhile, sheep and cattle, which do not occur at our study site, are considered grazers [45].

Riesch, et al. [14] found that red deer primarily forage on heathland in winter, thereby removing up to 59% of net primary productivity, corresponding to 0.45 animal units per hectare. Additionally, red deer maintain the heathland ecosystem by forming open sand structures and creating early stages of succession [49]. According to [15], free-ranging red deer remove as much as 14 kg N ha$^{-1}$ year$^{-1}$ from heathlands and it has been concluded that it makes them suitable for heathland conservation because red deer might remove enough nitrogen to cause nutrient depletion given, of course, that nitrogen deposition levels are sufficiently low. Removing red deer with exclosures promotes a general change in species composition, reduces bare soil occurrence, and increases the height of dwarf shrubs [37,49]. Findings from Smith, et al. [50] suggest that excluding large herbivores from heathlands increases carbon in the ecosystem, and their results disclose that the amount of carbon increased further in exclosures that received a substantial amount of atmospheric nitrogen (i.e., >11 kg N ha$^{-1}$ year$^{-1}$). A study from the United Kingdom emphasises that *C. vulgaris* bushes seem to spread by removing sheep from heathlands. However, this is not the case in areas with a high density of red deer, suggesting a considerable effect of wild grazers on vegetation composition on heathlands [51]. Also, Riesch, et al. [49] mentioned the importance of gathering evidence from deer grazing alone and combining it with other management treatments; however, their published work from 2020 did not include further treatments for heathlands.

In this study, we established a manipulation experiment on a Danish dune heath to study the effects of nitrogen deposition, mowing, and deer grazing on dune heath vegetation. The initial expectations were that (a) higher nitrogen deposition would amplify the effect of stress factors, thereby eventually decreasing the cover of dwarf shrubs; at the same time, more nitrogen would increase plant growth of dwarf shrubs, (b) nitrogen

deposition would increase graminoids' cover and plant growth, (c) more nitrogen would reduce the cover of bryophytes and lichens, (d) an increase in graminoid/dwarf shrub ratio would be observed with an increased dose of nitrogen given to the plots, (e) mowing would affect the dwarf shrub growth and cover negatively and promote graminoid growth and cover, and (f) species and species assemblies would have a higher cover and plant growth within the exclosure compared to plots outside exposed to grazing.

## 2. Materials and Methods

### 2.1. Site Description

The manipulation experiment was established on a 12 ha dune heath (code 2140: Decalcified fixed dunes with *Empetrum nigrum*) [1], Vust heath, in Northern Jutland, Denmark, 57°7′23.412″ N, 9°0′42.443″ E (Figure 1). The study site is within the NATURA 2000 area no. 16 Løgstør Bredning, Vejlerne og Bulbjerg. Historically, Vust heath has been part of a much larger coastal heath. However, extensive planting of spruce and pine plantations in the nineteenth and twentieth centuries has reduced the heath area considerably [52,53]. The flora is dominated by wavy hair grass *Avenella flexuosa* (L.) Drejer, *C. vulgaris*, *E. nigrum*, and areas with an extensive cover of lichens and bryophytes, particularly *Cladonia* spp., *Dicranum* spp., and red-stemmed feathermoss *Pleurozium schreberi* (Brid.) Mitt. Only a few non-native species were recorded within the experimental site, of which only heath star-moss *Campylopus introflexus* (Hedw.) Brid. was documented using the pinpoint method (a method for monitoring plant vegetation explained below). The north and east parts of Vust heath are adjacent to a pine forest that meets the sea five kilometres north of the study site. The climate is characterised by relatively wet winters and cold and wet summers, where the average yearly temperature and precipitation are 8.2 °C and 895.5 mm year$^{-1}$, respectively [54]. Nitrogen deposition was 9.43 kg N ha$^{-1}$ year$^{-1}$ in 2015 [55]. Soils are sandy Arenosol [56]. The average pH in the A-horizon was 3.79 (0.03 SE) ($n$ = 80), while the pH was 3.23 (0.02 SE) in the O-horizon ($n$ = 80). Soil pH was measured in $CaCl_2$, and soil samples were taken in 2020. In the A-horizon, the mean carbon/nitrogen ratio (C/N) was 26.5 ($n$ = 80), and the mean C/N in leaf tissue was 30.03 for *C. vulgaris* ($n$ = 14), 31.3 for *A. flexuosa* ($n$ = 26), and 33.31 for *E. nigrum* ($n$ = 31). Based on aerial photos from 1954 to 2022, we visually estimated a significant decline in the site's coverage of *C. vulgaris* and other dwarf shrubs. This decline in coverage is evident from the darker nuances signifying dwarf shrubs on the aerial photos being less disseminated in the area now than seventy years ago (Figures S4 and S5) [57]. Judging from the comprehensive data from NOVANA (a Danish habitat surveillance program) and our floristic surveys, we assessed most of the area as dune heath (Habitat type: 2140) [2,58]. Despite being a relatively small dune heath, the site had a variety of habitats and diverse flora. In general, many parts of the dune heath had several species of dwarf shrubs (Figure S17). In 2020, most *C. vulgaris* bushes were infested by *L. suturalis*, and data indicate they did not recover the following year; however, the results are non-significant (Figures S14 and S15).

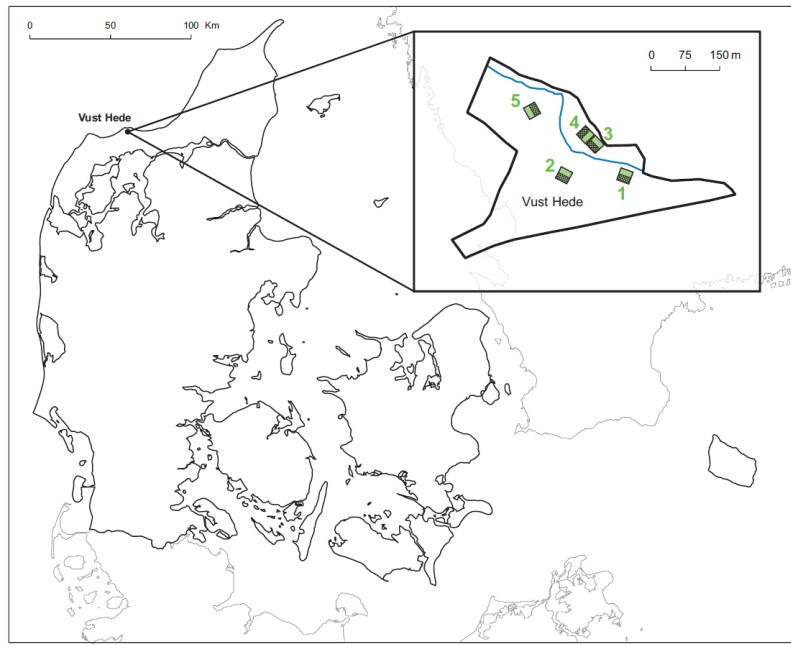

**Figure 1.** Map showing the location of Vust heath (Danish: Vust hede). Here, the rectangles are the Blocks (1–5), and the dark parts are the exclosure.

*2.2. Experimental Design*

Before establishing the experiment, we did not perform a baseline of the site's original vegetation and soil conditions. However, the first pinpoint sampling was taken in the autumn 2018 before the plots were mowed. The experiment was a full factorial experimental design with five blocks, each with 16 plots, where the manipulations of nitrogen and mowing were randomised within the two halves of the block. The experiment was similar to the experiment set up by [9] (Figure S7 and Figure 2). When established in 2018, each block was 26 m × 26 m, where half was fenced (exclosures). The fence was approximately two meters tall, keeping out the three deer species: red deer, fallow deer, and roe deer. The squared plots were (4 m × 4 m) separated by a 2 m buffer zone. Of the sixteen plots within a block, four received no nitrogen, four received 5 kg N ha$^{-1}$ year$^{-1}$, four received 10 kg N ha$^{-1}$ year$^{-1}$, and the last four received 25 kg N ha$^{-1}$ year$^{-1}$ to mimic realistic deposition levels. Nitrogen additions were applied four times yearly in April, May, June, and August to ensure nitrogen was added during growth. For the experiment, we used ammonium nitrate dissolved in water. Half the plots were mowed yearly in October ($n = 40$). Thus, post-mowed vegetation height was ca. 2–3 cm. All of the material was subsequently removed. We had a high mowing intensity because the project period only lasted a few years, but also to imitate the situation where high nitrogen deposition increases yearly biomass. Therefore, in the future, removing the biomass might be necessary relatively frequently to preserve the dune heath. In 2020 and 2021, we installed five game cameras to monitor which deer species occur at the site, game densities, and the time the different deer species spent foraging. We placed each camera in the centre of the block, overlooking the unfenced part of the block. The flora composition in block one (Figure 1) consisted primarily of native stress-tolerant graminoids and creeping willow *Salix repens* L. within the exclosure. In contrast, the plots outside the exclosure had a flora composition of the ericoid *E. nigrum* or the vesicular-arbuscular petty whin *Genista anglica* L., which has a symbiotic relationship with *Rhizobium leguminosarum* (Frank 1879) Frank 1889 Strandberg pers. comm.: Strandberg pers. comm.: [59,60]). Block two differed markedly from the others in that it had larger areas covered primarily by *C. vulgaris* and *G. anglica* outside the exclosure. In contrast, *A. flexuosa* and *C. arenaria* were dominant within some parts of the exclosures of blocks one and two. Block three had large areas covered with several species of lichens, e.g., *Cladonia portentosa* (Dufour) Coem. (1865) and *Cladonia ciliata* Stirt. In

contrast, *E. nigrum* in block four tended to dominate most plots. Block five was situated in a depression and relatively humid wet heath with a mixed cover of *C. vulgaris*, cross-leaved heath *Erica tetralix* L., bog-myrtle *Myrica gale* L., bog bilberry *Vaccinium uliginosum* L., and a diverse lichen flora (Figure S16).

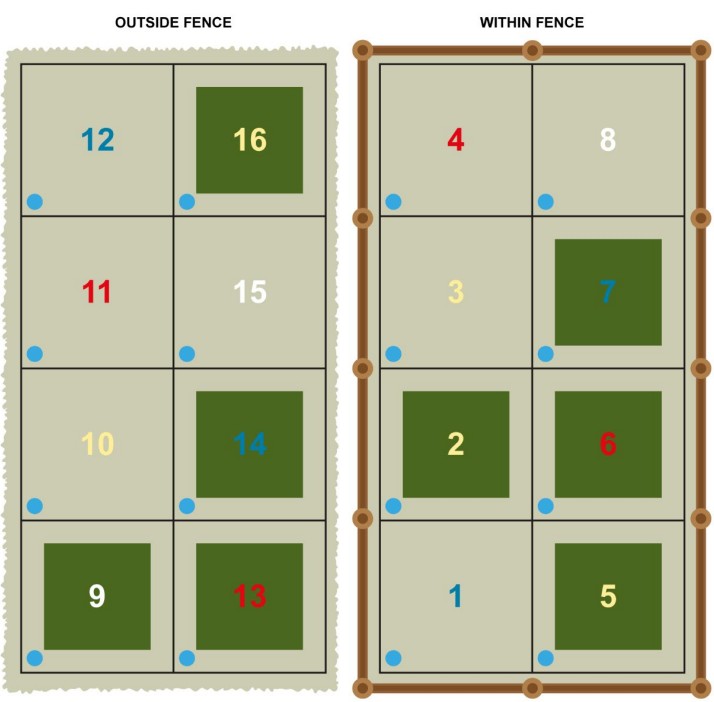

**Figure 2.** An example of a block's general structure. Green squares are mowed fields. Blue dots are fixed spots for the pinpoint frame. The numbers in different colours designate the given dose of nitrogen, with white being the control, blue being the low dose of 5 kg N ha$^{-1}$ year$^{-1}$, yellow being 10 kg N ha$^{-1}$ year$^{-1}$, and red being 25 kg N ha$^{-1}$ year$^{-1}$. The brown rectangle is the eight plots surrounded by the fence. All treatments were randomised.

### 2.3. Sampling and Data Handling

In this study, we used the pinpoint method, where a pinpoint frame is a metal frame with cords crossing each other (Figure S18). The position for taking pinpoint samples was fixed in each plot (marked with a blue dot in Figure 2). We lowered a pin 25 times where the cords crossed each other in the pinpoint frame to measure flora cover (i.e., if the pin hit a species at a cross or not). Ratio data were generated from the cover data, i.e., if there was a change in the cover of two species or species groups. Plant growth was measured as vertical density, i.e., the number of times the pin hit a specific species from the top of the vegetation to the soil surface at each cross [61]. Summer sampling was performed in early June (2019, 2020, and 2021), while autumn sampling was performed in August (2019 and 2021) and October (2018 and 2020). Data consisted of 2676 pin presence observations (Table S25). The recorded vascular plant and cryptogram species (where cryptogam individuals identified at the genus level were counted as a species) in all 80 plots were 23 and 10, respectively (Tables S23 and S25).

The data sets in the analyses comprised species aggregated into groups, where, for example, "Graminoids" includes the grass *A. flexuosa*, the sedge *C. arenaria*, sheep's fescue *Festuca ovina* L., the field wood-rush *Luzula campestris* (L.) DC., and mat grass *Nardus stricta* L. A defining characteristic of the dune heath flora is its dwarf shrub community. Therefore, we found it relevant to aggregate species into one group called "Dwarf shrubs". The group encompasses *C. vulgaris*, *E. nigrum*, *E. tetralix*, *G. anglica*, *M. gale*, and *S. repens*. All recorded lichens and bryophytes were grouped into "Lichens" or "Bryophytes", respectively. We identified some bryophyte specimens at the species level; however, no attempts were made

to compile a complete species list of lichens and bryophytes. Some individual species were subsequently considered in a separate analysis. These were *A. flexuosa, C. arenaria, C. vulgaris, E. nigrum,* and the bryophyte *P. schreberi.*

In 2019 and 2020, soil and leaf tissue samples were gathered from all plots, and in the mowed plots, the leaf tissue material was divided into seven species or groups and the dry weight was noted (Figure S2). Soil samples were collected using a portable soil sampler. Upon collection, the soil and leaf-tissue samples were stored at −18 °C. Before analysing the C/N ratio, samples were dried in paper bags at 60 °C for 48 h, and soil samples were crushed with a mortar and sieved with a 2 mm sieve. Weight measurements were noted. We took a subsample for the C/N measurements and crushed it on a Retsch Planetary Ball Mill Type PM 400, after which the C/N measurements were performed on a Thermo Flash EA 1112 nitrogen and carbon analyser. The sample was weighed in tin foil, around 50 mg, with three decimals accuracy. Hereafter, samples were transferred to the C/N apparatus and burned at 1800 centigrade. The combustion gasses were led through a GC column and separated into N and C, which a thermal conductivity detector detected. Organic leaf tissue samples were cut into smaller pieces and crushed on the Retsch Planetary Ball Mill, as were the soil samples. Nitrogen was measured in mg g$^{-1}$ dry material. Carbon was measured in mg g$^{-1}$ dry material, and the results were reported in percentage of dry weight, i.e., 1% = 10 mg/g dry weight. Soil pH was measured in 0.01 M CaCl$_2$·H$_2$O on a pH meter (PHM220) from Radiometer analytical$^®$ following this procedure: 5 mL of lump-free soil from the bleached sand layer was measured. After that, it was dried in a heating cabinet at 60 °C for 48 h. The sample was (1) poured into a glass, (2) dispensed with 25 mL of 0.01 M CaCl$_2$·H$_2$O, and (3) shaken vigorously for 5 min in a shaker. The suspension was ready for measurement after two hours of standing.

*2.4. Models*

The model for cover data was specified as a four-way mixed beta-binomial linear model with interactions and blocks and plot ID numbers as random components [62]. The code syntax is shown in Figure S3. Tables 1 and 2 show the models' output. In the article, mowing is specified as "mowing" or "harvest", and exclosure is specified as "fence" or "exclosure". The fixed effects were fence, dose, harvest, and year. Fence and harvest were specified as categorical variables: fence = "outside the fence" or "within the fence"; harvest = "yes" or "no" and dose and year as continuous variables, where dose = 0, 5, 10, and 25 kg N ha$^{-1}$ year$^{-1}$ and year = 1, ..., 4. The cover data were assumed to be beta-binomially distributed [63]. For vertical-density data analyses, we assumed that plant counts $y_i$ (i.e., number hits of individual plant hits) follow a negative-binomial distribution [64]. The model for the vertical density data was specified as a four-way mixed negative-binomial linear model. Otherwise, the model specification follows the one described for cover data. Models for the ratio data were fitted using a beta distribution in the package glmmTMB [65]. Also, the model specification follows the one described in the cover data analysis. The parametrisation follows the one in the package glmmTMB. Inspecting residuals from the cover models yielded a non-significant Kolmogorov–Smirnov dispersion and outlier test regarding all models, except autumn data from *E. nigrum*. The results of a log-likelihood ratio test on a Poisson model and a negative-binomial mixed model (NBMM) from the same vertical data supported the choice of the negative-binomial models. However, some NBMMs had a significant Kolmogorov–Smirnov test or dispersion test.

**Table 1.** Results from the analysis of the cover data. We fitted a model with all covariates and their interactions. The models are reduced using a log-likelihood ratio test, where the terms in the grey boxes are non-significant and not included in the final model, and the terms in the white boxes are included in the final model. The arrows indicate a significant positive effect (↑) or negative effect (↓) of the treatment. Stars indicate significant effects, where "***" is a *p*-value < 0.001, two stars "**" is a *p*-value < 0.01, and one star "*" is a *p*-value < 0.05. So, e.g., the final dwarf shrubs model from the summer data encompasses the covariates year, dose, fence, and harvest, shown as white boxes here.

| Models for Summer | Year | Dose | Fence | Harvest | Year:Dose | Year:Fence | Year:Harvest | Dose:Fence | Dose:Harvest | Fence:Harvest | Year:Dose:Fence |
|---|---|---|---|---|---|---|---|---|---|---|---|
| *Grasses* | | | | * ↓ | | | * ↑ | | | | |
| *Dwarf shrubs* | * ↓ | | | *** ↓ | | | | | | | |
| *Empetrum nigrum* | | | | * ↓ | | | *** ↓ | | | | |
| *Avenella flexuosa* | * ↑ | | | * ↓ | | | * ↑ | | | | |
| *Calluna vulgaris* | *** ↓ | | | ** ↓ | | | ** ↑ | | | | |
| *Carex arenaria* | | * ↑ | | | | | | * ↓ | | | * ↑ |
| *Lichens* | | | | | | | | | | ** ↓ | |
| *Bryophytes* | *** ↑ | | | | * ↓ | | | | | | |
| *Pleurozium schreberi* | *** ↑ | | | | | | | | | | |
| *Dicranum* spp. | *** ↑ | | | * ↑ | | | | | | | |
| **Models for autumn** | **Year** | **Dose** | **Fence** | **Harvest** | **Year:Dose** | **Year:Fence** | **Year:Harvest** | **Dose:Fence** | **Dose:Harvest** | **Fence:Harvest** | **Year:Dose:Fence** |
| *Grasses* | | | | | | | ** ↑ | | | | |
| *Dwarf shrubs* | | | | | | | *** ↓ | | | | |
| *Empetrum nigrum* | | | | | | | *** ↓ | | | | |
| *Avenella flexuosa* | | | | | | | ** ↑ | | | | |
| *Calluna vulgaris* | *** ↓ | | | | | | | | | | |
| *Carex arenaria* | | | | | | | | | | | * ↑ |
| *Lichens* | * ↓ | | | | | | * ↑ | | | ** ↓ | ** ↓ |
| *Bryophytes* | ** ↓ | | | | | | ** ↑ | | | | * ↓ |
| *Pleurozium schreberi* | * ↓ | | | | | | * ↑ | | * ↑ | | * ↓ |
| *Dicranum* spp. | | | | | | | ** ↑ | | | | |

**Table 2.** Results from the analysis of vertical density data. We fitted a model with all covariates and their interactions. The models are reduced using a log-likelihood ratio test, where the terms in the grey boxes are non-significant and not included in the final model, and the terms in the white boxes were included in the final model. The arrows indicate a significant positive effect (↑) or negative effect (↓) of the treatment. Stars indicate significant effects where "***" is a *p*-value < 0.001, two stars "**" is a *p*-value < 0.01, and one star "*" is a *p*-value < 0.05. So, e.g., the final dwarf shrubs model from the summer data encompasses the covariates year, dose, fence, and harvest, shown as white boxes here.

| Models for summer | Year | Dose | Fence | Harvest | Year:Dose | Year:Fence | Year:Harvest | Dose:Fence | Dose:Harvest | Fence:Harvest | Year:Dose:Fence |
|---|---|---|---|---|---|---|---|---|---|---|---|
| *Grasses* | | | * ↓ | | | | ** ↑ | | | | |
| *Dwarf shrubs* | *** ↓ | | | *** ↓ | | | | | | | |
| *Empetrum nigrum* | | | | * ↓ | | | *** ↓ | | | | |
| *Avenella flexuosa* | * ↑ | | | | | | ** ↑ | | | | |
| *Calluna vulgaris* | ** ↓ | | | | | | | | | | |
| *Carex arenaria* | | ** ↑ | | | | | | * ↓ | | | * ↑ |
| **Models for autumn** | **Year** | **Dose** | **Fence** | **Harvest** | **Year:Dose** | **Year:Fence** | **Year:Harvest** | **Dose:Fence** | **Dose:Harvest** | **Fence:Harvest** | **Year:Dose:Fence** |
| *Grasses* | | | | | | | *** ↑ | | | | |
| *Dwarf shrubs* | | | | | | | *** ↓ | | | | |
| *Empetrum nigrum* | | | | | | | *** ↓ | | | | |
| *Avenella flexuosa* | | | | | | | *** ↑ | | | | |
| *Calluna vulgaris* | | | | | | | | | | | |
| *Carex arenaria* | | | | | | ** ↑ | | | | | |

The model aimed to capture vegetation changes through the years; therefore, we focused on the interaction between year and fence, year and dose, year and harvest, and higher-order effects with years to track changes within the plots. A log-likelihood ratio test for model reduction was used in all models, correspondingly for the negative-binomial mixed models and the models for describing ratio data. Summer and autumn data were analysed separately; the results are shown in Tables 1 and 2. Graphs and figures were created using the program R and the "tidyverse" package [66,67]. The models were fitted in R with the packages lme4 and glmmTMB [65,68], and residual analysis was performed in DHARMa [69].

## 3. Results

Here, we present the results with significant higher-order effects containing the variable "year" from analyses of the pinpoint data. Significant effects are listed in Tables 1 and 2. Dwarf shrub cover and vertical density were not affected by excess nitrogen, nor were graminoid cover and vertical density. Inspecting the site's ortho and aerial photos, we visually assess that there has been a decline in the overall dwarf shrub cover at the site over the past seventy years (Figures S4 and S5). Excess nitrogen positively impacts the vertical density and coverage of *C. arenaria*. Bryophytes and the bryophyte species *P. schreberi* responded negatively to higher doses of added nitrogen in autumn (Figure 3 and Table 1). In addition, the interaction between year and dose was significantly negative for bryophytes in summer. We refrain from concluding anything on trends in lichens and other bryophyte species because we assessed that the sample size here was too small; however, the results from *Dicranum* spp. and lichens are shown in Tables 1 and 2. The ratio data did not display any significant terms. The results from biomass mowing give a linear correlation between vertical density and biomass for graminoids (Figure S1).

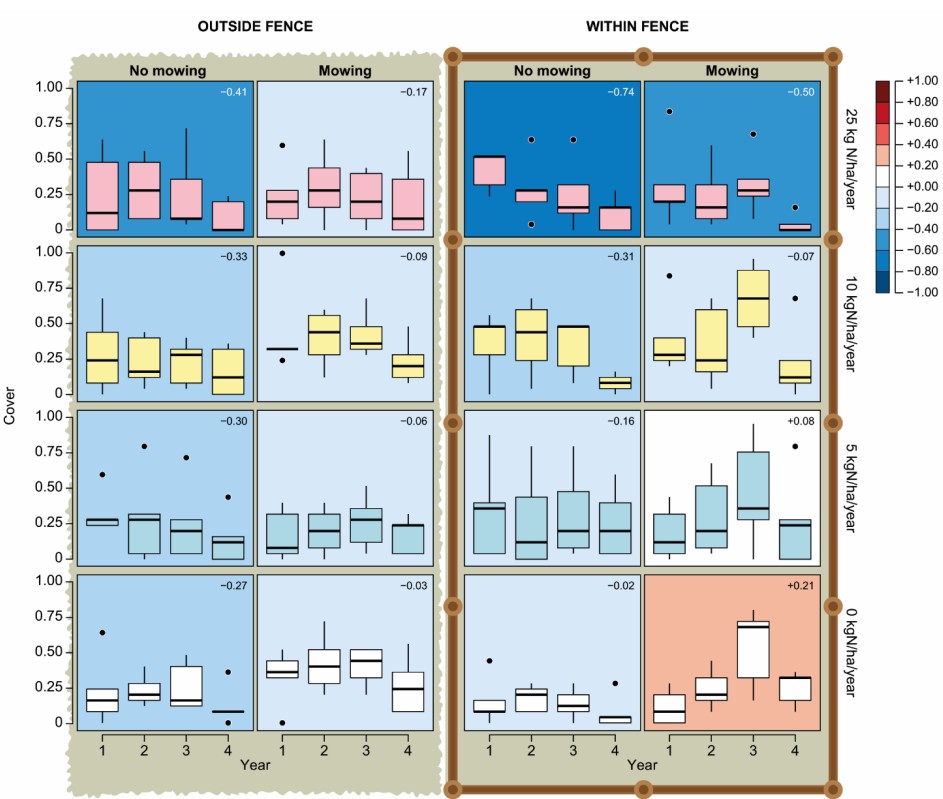

**Figure 3.** The calculated slope of the cover of bryophytes in autumn and the raw data presented in boxplots. The average slope is in the top right corner of each plot, with red being a positive slope and blue a negative average slope. The brown rectangle on the right side of the graph is the fence surrounding eight plots (i.e., plots without deer grazing). The different colours of the boxplots refer to the given dose, where the red boxplots designate a high dose of nitrogen added to the plots (25 kg N ha$^{-1}$ year$^{-1}$), yellow boxplots designate a dose of 10 kg N ha$^{-1}$ year$^{-1}$, and blue boxplots are a low dose with 5 kg N ha$^{-1}$ year$^{-1}$, while white is the control, or 0 kg N ha$^{-1}$ year$^{-1}$. The legend indicates the value of the average slope shown to the right.

The dwarf shrub coverage fell during the four years within the mowed plots in autumn, i.e., there was a negative effect of the interaction between year and mowing. Additionally, *E. nigrum* declined significantly in autumn and summer over the four years in the mowed plots. The cover of *C. vulgaris* decreased during the four years. Data from *C. vulgaris* display an apparent positive effect of the interaction between year and mowing in summer. Based

on the autumn and summer data, the cover of graminoids within the mowed plots had an increased coverage during the experiment (Figure 4 and Figure S8). For the graminoids' autumn and summer data, the vertical density increased the most at the mowed plots (Figures S12 and S13). The grass *A. flexuosa*, increased in cover within the mowed plots over the four years, both for the summer and autumn data (Tables S1 and S2).

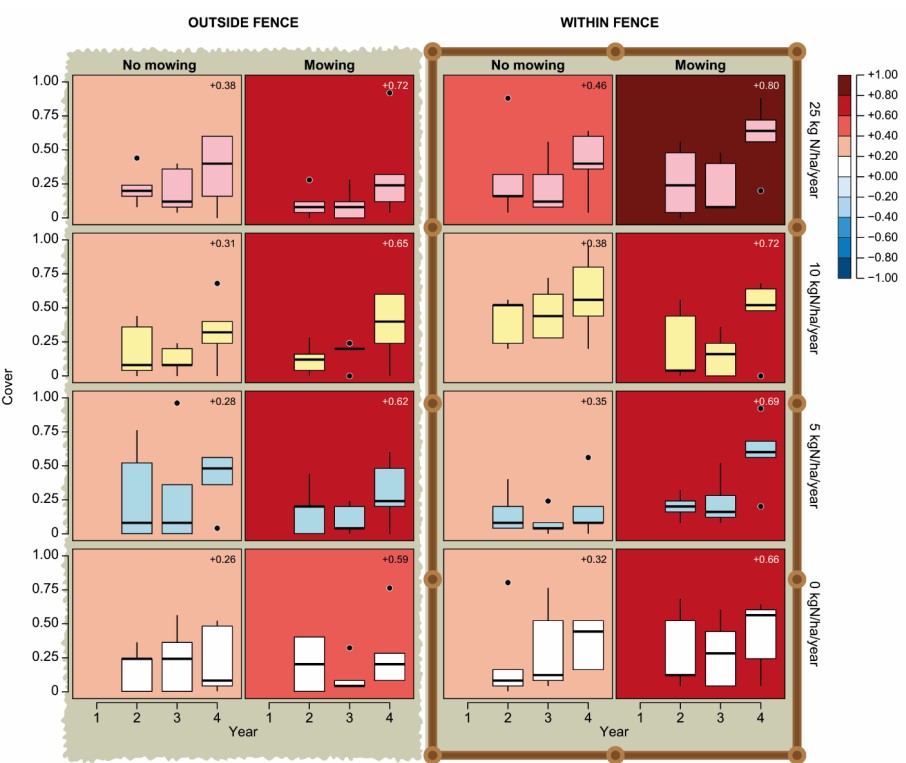

**Figure 4.** The calculated slope of the cover of graminoids in summer and the raw data presented in boxplots. The average slope is in the top right corner of each plot, with red being a positive slope and blue a negative average slope. The brown rectangle on the right side of the graph is the fence surrounding eight plots (i.e., plots without deer grazing). The different colours of the boxplots refer to the given dose, where the red boxplots designate a high dose of nitrogen added to the plots (25 kg N ha$^{-1}$ year$^{-1}$), yellow boxplots designate a dose of 10 kg N ha$^{-1}$ year$^{-1}$, and blue boxplots are a low dose with 5 kg N ha$^{-1}$ year$^{-1}$, while white is the control, or 0 kg N ha$^{-1}$ year$^{-1}$. The legend indicates the value of the average slope shown to the right.

There was no effect of exclosure on dwarf shrubs or graminoids, except for the species *C. arenaria*. *Carex arenaria* increased cover and vertical density inside the exclosure and declined in plots with deer grazing in both periods (Figures S9 and S10). The effect was most substantial within the exclosure at non-mowed plots in autumn. Generally, bryophyte cover decreased more within the exclosure than outside in autumn. The bryophyte species most frequently recorded at the site was *P. schreberi*. It followed the same pattern as the aggregated group "Bryophytes".

It was not possible to estimate deer densities based on the camera data. However, the average foraging time was 58.5 s for roe deer (SE = 8.6); for red deer, the figure was 24.0 s (SE = 13.3) and for fallow deer was 77.3 s (SE = 21.0). The maximum number of deer counted in one photo (fallow deer) was nine, recorded during spring. In contrast, red deer had the lowest maximum number of deer counted in one photo (*n* = 2), and the species was not recorded during winter.

## 4. Discussion

The general decline in dwarf shrub cover at our site aligns well with reports from the Netherlands and elsewhere e.g., [16,31,70]. The cause of the decline is nitrogen deposition, which

enhances the effects of drought and frost and increases the abundance of *L. suturalis* [8,9,31]. Although we did not observe a significant effect of the outbreak of *L. suturalis* in 2020, the attack was visually extensive. Also, it seems like the vertical density of *C. vulgaris* was lower in non-mowed plots after the outbreak (Figures S14 and S15). If the event is recurrent, it might challenge the continuing characteristic *C. vulgaris* vegetation at the site. The non-significant effects of nitrogen on *C. vulgaris*, dwarf shrubs, and *E. nigrum* within the non-mowed plots could indicate an unchanged resistant ecosystem, at least in the short term, which implies that the dwarf shrub community might persist even at a high deposition level, unlike bryophytes, which are sensitive according to our data and data from others [71]. The results here differ from [72], which reported an increase in *E. nigrum* cover with more N and did not show a decline in *C. vulgaris* cover. Our analysis supports the results presented by [32], who assert that short-term nitrogen manipulation experiments do not affect the competition between dwarf shrubs and graminoids on *E. nigrum*-dominated heaths such as our dune heath. However, in the long-term, projected future nitrogen deposition levels and climate change, which enhances stress factors, might reduce resistance, challenging the dwarf shrub community on dune heaths and heathlands in general [73]. The sedge *C. arenaria* was positively affected by excess nitrogen, as documented by [35]. The positive effect indicates that even low chronic doses of nitrogen facilitate this species' expansion on decalcified dunes. Our study shows that a minor increase in nitrogen deposition decreases the cover of bryophytes. Bähring, et al. [9] made a comparable observation. Based on our data, the negative response of bryophyte cover is highest at plots with high doses of added nitrogen, which is likewise supported by [9]. It is essential to state that different species of bryophytes respond differently to nitrogen deposition and that, just as with vascular plants, there can be a different tolerance for nitrogen between species. The three most frequently occurring bryophytes in our data set are *Dicranum* spp., *Hypnum* spp., and *P. schreberi*, where at least one species in the genus *Dicranum* is more tolerant to nitrogen than *P. schreberi* [74]. The results from a calcareous grassland show that the calcifuge broom fork-moss *Dicranum scoparium* Hedw., probably the most common *Dicranum* species at our site, increases its frequency under more acidic conditions [75]. However, because there was little data, we did not have the opportunity to assess whether the *Dicranum* sp. increases or decreases in coverage. The significant reduction in coverage of bryophytes might be related to a higher canopy height and shoot extension of *C. vulgaris* in plots with high amounts of added nitrogen, consequently excluding more light; however, as stated above, our data and empirical analysis do not find any enhanced growth of *C. vulgaris* or other dwarf shrubs with increasing N, unlike [76], which reported an increased height and shoot extension of *C. vulgaris,* causing a decline in lichens and bryophytes. However, graminoids' increased cover and vertical density could explain the decline of bryophytes, as cryptogams are less competitive [9]. The lack of change in the ratio (results not shown) between dwarf shrubs and graminoids might be due to the mature, dense vegetation of dwarf shrubs, where dwarf shrubs win the competition [77,78].

Mowing removes accumulated nitrogen in the system, while an increased nitrogen deposition results in higher plant growth and more litter, meaning frequent mowing to remove excess nitrogen might be necessary [11,24]. However, our results suggest a decline in the dwarf shrub community using intensive annual mowing. As expected, graminoids seem to be favoured by intensive mowing and may dominate in the coming years, a result supported by [32]. Power, et al. [24] underline that mowing might not counteract the effects of higher nitrogen loads. Nevertheless, high-intensity mowing seems to mitigate some of the adverse effects of excess nitrogen on bryophytes. Suppose a high mowing frequency becomes required; the result may be that the dune heath turns into a grass-dominated habitat. In that case, other management forms, e.g., sod-cutting, might be required to prevent the dune heath from converting into grasslands [24].

The study indicates that grazing pressure from deer alone affects vegetation and corroborates that excluding the three deer species from dune heaths would change the characteristic heath vegetation. Our results showed that in the absence of deer, *C. arenaria*

increased in coverage, while deer slowed the decline of bryophytes in plots with high doses of added N. This supports the proposition that the three deer species, independently or combined, positively affect dune heath vegetation [49,51]. Our study is different from most other studies treating the effects of deer on heathlands because fallow deer was the most abundant deer species at our site, followed by roe and red deer. However, we assess red and fallow deer's effect on vegetation to be comparable because they are both intermediate grazers.

At the study site, there is no dung removal, which contrasts with traditional heathland farming practices, where livestock dung was removed and depleted the soil of nutrients; albeit, even when livestock is not present at our study site, deer are still exporting nutrients from the dune heath [15]. However, our data show that even with deer grazing, nitrogen-sensitive bryophytes declined. We conclude that deer density was too low to affect dwarf shrub cover adversely. Additionally, the lack of effects of deer grazing on the dwarf shrub cover could indicate a lower grazing pressure at our site than at similar sites, such as ones in Scotland [48]. There is no effect on the vertical density and cover of excluding deer on *E. nigrum*. One possible explanation might be the small hoof size of the three deer species, leaving the *E. nigrum* cover relatively intact.

Our study confirms the value of considering the combined effects of nitrogen deposition, mowing, and deer grazing because these drivers influence dune heaths separately and combined. The results document that the dwarf shrub vegetation resists increased nitrogen deposition, even at relatively high doses (i.e., 25 kg N ha$^{-1}$ year$^{-1}$). We highlight the adverse effects of mowing on dwarf shrubs. Notably, annual mowing seems to facilitate the expansion of graminoids. We welcome further studies on the effects of deer on heathlands and dune heaths at different grazing intensities to monitor the effects on dune heath vegetation. Further, we recommend removing trees and bushes from dune heaths, which is essential to prevent the areas from being overgrown, as this seems to be a typical problem.

Based on the results of our study, here are our general recommendations:

1. Bryophytes seem to be affected adversely even at deposition levels at or below the empirical critical load of dune heaths, which is 10 to 15 kg N ha$^{-1}$ year$^{-1}$. The decline of bryophytes seems to happen even in plots with annual mowing where the material is removed and with a low dose of added nitrogen (i.e., 5 kg N ha$^{-1}$ year$^{-1}$). Therefore, keeping nitrogen deposition levels low is pivotal to protect these sensitive elements.
2. It is essential to mention that our mowing method positively affects bryophytes and the moss species *P. schreberi*. However, we advise managers to avoid annual mowing near the soil surface because it promotes graminoids and causes a decline in *E. nigrum*, but not to avoid mowing in general.
3. Excluding deer enhances growth and increases coverage of *C. arenaria* at higher N deposition. Therefore, allowing deer numbers to increase might be beneficial for dune heaths.

**Supplementary Materials:** The following supporting information can be downloaded at: https://www.mdpi.com/article/10.3390/ecologies5010008/s1.

**Author Contributions:** M.E.K.: Conceptualisation, Formal Analysis, Investigation, Data Curation, Writing—Original Draft, Writing—Review and Editing, Visualisation. F.H.: Data Curation. J.L.B.: Conceptualisation, Methodology, Funding Acquisition, Project Administration. C.F.D.: Methodology, Software, Writing—Review and Editing, Supervision, Validation. M.T.S.: Data Curation. All authors have read and agreed to the published version of the manuscript.

**Funding:** The Aage V. Jensen Charity Foundation supported the project substantially; without their financial support, the project would not have been possible.

**Institutional Review Board Statement:** Not applicable.

**Informed Consent Statement:** Not applicable.

**Data Availability Statement:** The original contributions presented in the study are included in the article and Supplementary Materials, further inquiries can be directed to the corresponding author.

**Acknowledgments:** Lise Lauridsen and all laboratory technicians at the institute are thanked for their long fieldwork hours in Thy. Charlotte Elisabeth Kler is thanked for her valuable comments on the text in the manuscript's final and first drafts. Knud Erik Nielsen is thanked for his help with setting up the experiment. We also want to acknowledge Lasse Dümke for his help with running the experiment.

**Conflicts of Interest:** All authors declare no competing interests.

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
