# Peer review of "How Do Nitrogen Deposition, Mowing, and Deer Grazing Drive Vegetation Changes on Dune Heaths?"

_2673-4133, doi:10.3390/ecologies5010008_

Round 1
Reviewer 1 Report
Comments and Suggestions for Authors
A paper reporting an interesting applied research, on a sound scientific ground. To my view, it is well organized and exposed, and thus recomended for publication. Although the writting is to me very fine, I put a few marks and comments along the manuscript, to be considered.

Author Response
Thank you for thoroughly reading our article. Your suggestions are valuable, and most are incorporated in the final version. Your can check our comments directly in the pdf-file.
Reviewer 2 Report
Comments and Suggestions for Authors
The manuscript by Kaae et al., entitled “Nitrogen deposition, mowing, and deer grazing drive vegetation changes on dune heaths” is aimed at investigating the effects of nitrogen deposition mowing and deer grazing on different types of plant communities occurring on decalcified fixed coastal dunes in Northern Jutland. In general, the work is interesting because it deals with a rare and threatened ecosystem considered of European priority importance. The experimental work was done properly even if the methods are not always clear to me, and the results are not always shown in an appropriate way and should be discussed more in depth. In addition, some parts of the manuscript do not read well because the order of tables and figures is not always well arranged.
Considering all the above, I think that this manuscript is suitable for publication in MDPI Ecologies once the Authors will have addressed the issues listed below.
Specific comments
Title
Maybe, it would be better to formulate the sentence in an interrogative form.
Key words
As keywords are used for indexing, I think that words already present in the title should not be listed in the keywords. Please, replace “dune heats”, “Nitrogen deposition”. “Deer grazing” and “Mowing” ” with other keywords.
Introduction
Lines 34-38: when discussing the importance of dune heaths for conservation purposes, I think that it should be mentioned the fact that, according to Habitats Directive (Directive 92/43/CEE) this is an habitat of priority importance. Furthermore, in order to support the importance of the research, it would be useful to mention its conservation status according to the last report ex art. 17.
Lines 43-46: please, check the syntax.
Materials and Methods
Site description
Please, specify whether the study area falls within a Natura 2000 Network site and, if yes, which one (code and name).
Lines 201-208: I suggest summarising the information regarding the site into a table. Moreover, it could be useful to specify the distance from the sea.
Experimental design
Lines 249-256: these are more Results than Materials and Methods: please, move these lines to the Results section. I am not sure that I understood the way the average foraging times are presented: do the values refer to the time spent foraging per visit to the site? In the way they are currently presented, I understood that roe deer, for example, spend 58,5 seconds per year foraging at that site (but I do not think this is the case). Please, rewrite these lines to make the concept expressed clearer.
Sampling and data handling
I suggest to briefly explain the pinpoint method as it is not so commonly used, at least in Europe.
Results
Line 390: Figure 4 is mentioned in the text before Fig. 3, Also, Figure 4 is before Figure 3, Please, check this and change accordingly
A figure showing the calculated slope of the vertical density of Calluna vulgaris is not included in Appendix A. It would have been useful to compare Calluna with Empetrum. Please, add it.
Discussion
In the last paragraph of the Abstract, some considerations and suggestions about conservation management are reported, but there is no mention of them in the Discussion. Please, check this and change the text accordingly.
Lines 438-441: the authors state that the dwarf shrub vegetation cover has suffered a general decline. It is supposed that they have deduced this fact by comparing a recent orthophoto with the aerial photo of 1954 (Figures A4 and A5). They also state which are the causes of the decline according to the literature but their research doesn’t support this statement. I think that they should better justify the discordance of their results with those reported in the literature.
Lines 445-450: the authors state that there are non-significant effects of different nitrogen levels on dwarf shrubs. As far as I know of, Calluna vulgaris and Empetrum nigrum respond differently to excess Nitrogen as Calluna vulgaris is negatively affected while Empetrum derives a small benefit from it, as also demonstrated by figure A11. Therefore, I believe it would be useful to analyse the different response between the two species and if no differences are found, it would be appropriate to specify it.
Finally, Authors could give suggestions on the conservation management of this habitat, and suggestions for defining conservation measures as required by the Directive itself.
Appendix A
Models
Line 17, line 21: Please, replace Deschampsia flexuosa with Avenella flexuosa
Table A 23: Please, replace Deschampsia flexuosa with Avenella flexuosa
Table A 25: Please, replace Deschampsia flexuosa with Avenella flexuosa
Figure A7: line 83: squares are not visible.
